# Bayesian mediation analysis using patient-reported outcomes from AI chatbots to infer causal pathways in clinical trials

**Shihao Shen**[1], **Jun Yin**[2]*

**1** Agios Pharmaceuticals, Cambridge, Massachusetts, United States of America, **2** Moffitt Cancer Center, Department of Biostatistics and Bioinformatics, Tampa, Florida, United States of America

* vivien.yin@moffitt.org

## Abstract

The integration of artificial intelligence (AI) chatbots into clinical trials offers a transformative approach to collecting patient-reported outcomes (PROs). Despite the increasing use of AI chatbots for real-time, interactive data gathering, systematic frameworks for analyzing these rich datasets—especially in uncovering causal relationships—remain limited. This study addresses this gap by applying a Bayesian mediation framework to PROs collected via AI chatbot interactions, uncovering causal pathways linking treatment effects to outcomes through mediators like adverse events and patient-specific covariates. Using a simulation-based approach with GPT-4o, synthetic patient-chatbot dialogues were generated to evaluate the performance of the Bayesian mediation framework, which effectively decomposed total effects into direct and indirect components while quantifying uncertainty through credible intervals. The results demonstrated low bias (<0.05), robust coverage (>85%), in estimation of the direct, indirect effect and other variables of the mediation pathways, underscoring its potential to improve clinical trial data accuracy and depth. By integrating AI chatbot-based PRO collection with Bayesian mediation analysis, this study presents a scalable and adaptive framework for quantifying causal pathways, enhancing the quality of patient-reported data, and supporting personalized, data-driven decision-making in clinical trials.

## Introduction

Artificial intelligence (AI) chatbots is being studied as a scalable and interactive method for collecting patient-reported outcomes (PROs) in clinical trials, providing real-time monitoring and personalized patient engagement. For instance, in an oncology setting in Brazil, an AI-based chatbot initiated the majority of patient dialogues, 97% of 3,883 conversations, facilitating early detection of adverse events and reinforcing adherence to treatment [1]. In diabetic retinopathy research, the PROBot

**Data availability statement:** All simulation data are available from the Github https://github.com/SSDataTalk/PROdata.

**Funding:** The author(s) received no specific funding for this work.

**Competing interests:** The authors have declared that no competing interests exist.

framework harnesses large language models to capture enhanced PRO measures, helping improve compliance and reduce vision impairment [2]. Similarly, "Lucy LiverBot", deployed in Australia for patients with decompensated cirrhosis, demonstrated not only an increase in health literacy but also unforeseen benefits for mental well-being [3]. These studies underscore the potential of AI chatbots to refine data collection, bolster patient engagement, and ultimately advance clinical trial outcomes.

In addition to improving data collection in clinical trials, AI chatbots confer several other benefits by facilitating real-time, interactive patient-reported outcomes (PROs) [4–6]. Their user-friendly format promotes consistent symptom and side-effect reporting, while built-in automation enables scalability across large patient populations and real-time responses to emerging health needs. These tools also enhance data accuracy by reducing human error, offer greater convenience and accessibility—especially for individuals with limited mobility or living in remote areas—and can significantly lower operational costs by minimizing manual data collection. Further, AI-driven analysis of patient data detects early warning signs and generates actionable insights, advancing personalized and patient-centered care by delivering targeted education and emotional support throughout treatment.

AI chatbots enhance insights into the causal pathways leading to PROs by leveraging their capabilities in data collection, analysis, and contextual understanding [7]. By capturing both structured responses (e.g., symptom severity and timing) and unstructured data in the form of patient narratives or nuanced symptom descriptions, these tools build a rich, holistic dataset for uncovering cause-and-effect dynamics. Their real-time, longitudinal tracking of patient responses further enables the detection of evolving symptom patterns, while adaptive questioning drills deeper into potential contributing factors—such as lifestyle, environment, or psychological states—that traditional surveys might overlook. Advanced machine learning algorithms then integrate these multifaceted data streams, identifying subtle correlations and mind-body interactions (e.g., stress-related gastrointestinal symptoms) with predictive modeling that can inform personalized interventions. When deployed at scale, chatbots aggregate data across diverse populations, clarifying common causal pathways, subgroup-specific variations, and the relative effectiveness of different interventions. Altogether, these capabilities empower researchers and clinicians to generate a more complete and actionable understanding of the complex forces that drive health outcomes.

Bayesian mediation models play a pivotal role in elucidating the pathways through which an exposure exerts its influence on an outcome, especially when the goal is to separate total effects into direct and indirect components [8,9]. Unlike their frequentist counterparts, these models incorporate prior information and generate posterior distributions for each parameter, enabling more nuanced inferences about mediator and exposure relationships. By quantifying uncertainty in a probabilistic framework, Bayesian methods allow researchers to better account for sampling variability and model complexity—particularly when dealing with small sample sizes, missing data, or intricate mediation structures. This approach provides a richer, more flexible

platform for integrating new and existing evidence, thus enhancing the precision and interpretability of causal inferences in clinical trials and other complex healthcare studies.

Here, we leverage patient-reported outcomes gathered through an AI chatbot to fit a Bayesian mediation model aimed at uncovering the causal pathways linking the exposure to the outcome. By incorporating both structured and unstructured data from chatbot interactions—such as symptom severity, behavioral changes, and nuanced patient narratives—into our modeling, we gain a more granular view of how indirect effects operate through mediators. This Bayesian approach allows us to integrate prior information and continuously update our inferences as more data becomes available, a key advantage when investigating complex clinical scenarios. Moreover, the probabilistic framework of Bayesian mediation enables a more transparent quantification of uncertainty, particularly beneficial when analyzing large-scale, real-world data from AI-driven PRO collection in clinical trials.

The objective of this study is to evaluate the application of a Bayesian mediation framework to analyze PROs collected through AI chatbot interactions in a clinical research setting. Specifically, we seek to address the research questions of (1) whether the Bayesian mediation framework can reliably estimate direct and indirect effects using PRO data gathered from chatbot conversations, and (2) we aim to demonstrate how causal pathways between treatment exposure and patient outcomes can be uncovered using chatbot-collected data. In this framework, the primary exposure variable is the treatment assignment (active vs. placebo), the primary outcome is patient-reported fatigue, and the mediator is the occurrence of adverse events as reported during the chatbot dialogue. Additionally, stress-related factors reported by patients are incorporated as covariates influencing the outcome. By modeling these relationships, we seek to provide a novel and effective strategy for leveraging AI-driven PRO data to enhance causal inference and refine understanding of treatment effects in clinical trials.

## Materials and methods

We design an AI chatbot to gather the PRO related to fatigue, stress and adverse effect, using a structured conversation flow illustrated in Fig 1. The chatbot begins by asking how the user feels, then specifically inquires whether the user has experienced fatigue. If the answer is "Yes," it explores possible social or cultural reasons for the fatigue before delving into whether treatment side effects might be causing it. On the other hand, if the user reports "No" fatigue, the chatbot asks about social or cultural stress and subsequently checks for any side effects of the treatment. This branching logic tailors follow-up questions based on the user's initial response, guiding the conversation toward identifying potential psychological, social, or medical factors related to the user's condition.

To simulate chatbot–user interactions, we used GPT-4o [10] to alternate between the roles of "agent" (the AI chatbot) and "user" (the patient), allowing generation of realistic, bidirectional conversations. We began by defining a set of representative patient concerns related to fatigue, stress, and side effects. These scenarios guided the chatbot's scripted decision logic, which was implemented through GPT-4o prompts to simulate realistic conversational pathways.. Then, we

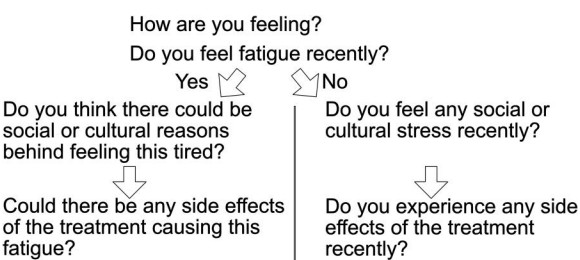

**Fig 1. Router of the chatbot to gather the PRO of fatigue, stress and adverse effect.**

implemented a scripted branching logic—mirroring the conversation flow depicted in our decision-tree diagram—to guide the generation of GPT-4 responses at each user input. Specifically, we employed role-based prompting in which GPT-4 was instructed to assume the "agent" role, formulating questions according to predefined conversational steps, and then switched to the "user" role to supply plausible patient statements or acknowledgments. Through iterative refinement, we ensured that the simulated dialogues accurately reflected the complexity and variability of genuine patient–chatbot interactions, including follow-up questions and context-sensitive inquiries. This approach generated a robust, synthetic dataset of conversation transcripts, which was later used to test and validate our Bayesian mediation model, particularly in understanding how different types of user feedback (e.g., admitting fatigue vs. not) might shape causal pathways in patient-reported outcomes.

As shown in Fig 2, we applied a Bayesian mediation framework to quantify the direct effect of treatment (X) on fatigue (Y), and the indirect effect mediated by adverse events (M), with stress (Z) included as a covariate influencing the outcome. Conceptually, X influences M (path a), which in turn affects Y (path b), and X may also exert a residual direct effect on Y (path c′). Additionally, Z enters the model as a predictor of Y (path d). In practice, we place prior distributions (often weakly informative normal priors) on each path coefficient and use Markov chain Monte Carlo (MCMC) sampling in PyMC [11] to estimate the joint posterior distributions of the model parameters. This allows us to derive credible intervals for the indirect effect (i.e., the product a×b) and the direct effect (c′) while propagating uncertainty throughout. The result is a posterior-based estimate of whether and how the treatment (X) influences fatigue (Y) both directly and through the mediator (M), taking into account the role of stress (Z).

Using definitions from Hayes (2018) [12], the effect of an exposure (X) on an outcome (Y) can be decomposed into three key components: the direct effect (c′), capturing the influence of X on Y independent of the mediator (M); the indirect effect (a·b), reflecting how X affects Y through the mediator; and the total effect (c = c′ + a·b), which is the sum of both paths. In a Bayesian mediation framework, each path coefficient is assigned a prior, and posterior distributions of c′, a·b, and c are derived via Markov chain Monte Carlo (MCMC) sampling. The posterior estimates, along with their credible intervals, allow researchers to ascertain not only whether mediation is present, but also the relative magnitudes of direct and indirect pathways, thereby providing a nuanced and probabilistic view of how X influences Y through M.

Based on the parametric Bayesian mediation framework, mediator model is

$M_i \sim \text{Normal}(i_M + a \cdot X_i, \sigma_M)$ And the outcome model is

$$y_i \sim \text{Normal}(i_y + c' \cdot X_i + d \cdot Z_i + b \cdot M_i, \sigma_Y)$$

where i indexes each observation, $i_M$ and $i_y$ are intercept parameters. Note that $X_i$, $Z_i$, $M_i$ and $Y_i$ are observed data.

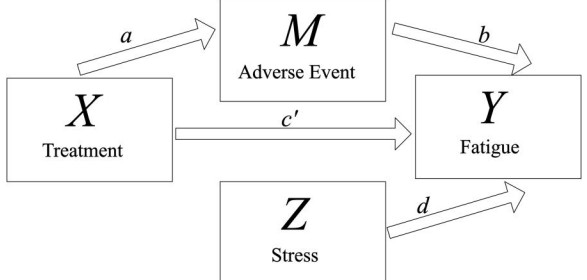

**Fig 2. Bayesian mediation framework.**

The direct effect given by $c'$ is the portion of the effect of the independent variable on the outcome that is not mediated by the mediator. The indirect effect given by $a \cdot b$ is the portion of the effect that is mediated through the mediator. The total effect given by $c' + a \cdot b$ is the sum of the direct and indirect effects.

## Results

We conducted 100 iterative simulations of chatbot–user dialogues using GPT-4o. In each simulation, 75 subjects were randomized 1:1 into active or placebo groups, defining their X variables. From each subject's dialogue, we extracted the values for M, Y, and Z, based on the simulated responses. Fig 3 provides an example of a single simulated chat, featuring a subject who reports fatigue, domestic stress, and an adverse event.

In our simulation result (Table 1), **a** represents the effect of the exposure (*X*) on the mediator (*M*), while **b** captures how changes in *M* translate into changes in the outcome (*Y*). The parameter **c′** (often referred to as the "direct effect")

**Chatbot:** Hi there, how are you feeling today? Have you been experiencing any fatigue recently?

**Subject:** Hello. Yes, I've actually been feeling pretty exhausted for the past few days.

**Chatbot:** I see. Do you think there could be social or domestic reasons that might be contributing to how tired you feel?

**Subject:** Possibly. I've been juggling a lot at home—keeping up with my kids' activities, plus managing some financial stress. It's been overwhelming.

**Chatbot:** That sounds challenging. Do you think there could also be any side effects from your current treatment or medication that might be causing or adding to this fatigue?

**Subject:** It's something I'm starting to wonder about. It feels like I've had worse insomnia than usual.

**Chatbot:** Understood. It might be a good idea to speak with your healthcare provider about these concerns. They can help rule out or confirm if the new treatment is contributing to your fatigue and explore possible solutions.

**Subject:** Definitely—I'll schedule an appointment and see if there's anything we can adjust.

**Chatbot:** That's a great plan. Meanwhile, make sure to take care of yourself, both mentally and physically. If you feel your stress levels rising, try to reach out for support or consider simple stress-relief activities.

**Subject:** Thank you. I really appreciate the advice. It helps to know there are steps I can take to figure this out.

**Fig 3. An example of the simulated chat.**

**Table 1. MCMC simulation results.**

|  | Bias | RMSE | Coverage | Interval_Width |
|---|---|---|---|---|
| a | 0.00397 | 0.027764 | 0.97 | 0.13721 |
| b | 0.0264 | 0.248215 | 0.91 | 0.81048 |
| cprime | 0.00303 | 0.105357 | 0.97 | 0.42059 |
| d | −0.00146 | 0.113848 | 0.94 | 0.42598 |
| im | −0.03691 | 0.04929 | 0.87 | 0.10089 |
| iy | 0.06435 | 0.248244 | 0.87 | 0.81146 |
| σm | −0.00444 | 0.041313 | 0.5 | 0.04341 |
| σy | 0.00832 | 0.029047 | 0.95 | 0.10416 |
| indirect effect | 0.00214 | 0.009217 | 1 | 0.04909 |
| total effect | −0.11428 | 0.184343 | 0.86 | 0.59791 |

measures the influence of *X* on *Y* that is not channeled through *M*. We include **d** as a coefficient for an additional covariate—such as stress—that can also predict *Y*. Meanwhile, **im** and **iy** are intercepts for the mediator and outcome equations, respectively, indicating baseline levels of *M* and *Y*. The parameters **σm** and **σy** represent the residual (or unexplained) variability in *M* and *Y*, respectively. Finally, the **indirect effect** (a × b) quantifies the portion of *X*'s influence on *Y* transmitted through *M*, whereas the **total effect** is the sum of the indirect effect and the direct effect (*c′*), providing an overall measure of *X*'s impact on *Y*.

In our simulation outputs, **Bias** represents the average deviation of the estimated parameter from its true value, indicating whether the estimates tend to systematically over- or underestimate the parameter. **RMSE** (root mean square error) reflects both the variance of the estimator and its bias, offering a single metric for the overall accuracy of the estimate. **Coverage** denotes the proportion of simulated datasets for which the 95% credible interval (CI) includes the true value, thus capturing how well our interval estimates align with the real parameter. **Interval_Width** is the mean width of those CIs, indicating how precise (narrow) or imprecise (wide) the intervals are.

Our simulation results indicate generally low bias and acceptable root mean square error (RMSE) for the primary path coefficients (*a*, *b*, *c′*, and *d*), with coverage values approaching or exceeding the conventional 0.90–0.95 range. For instance, *a* and *c′* exhibit minimal bias (0.00397 and 0.00303, respectively), robust coverage (0.97), and relatively narrow interval widths. Meanwhile, *b* shows a larger RMSE (0.248) and somewhat lower coverage (0.91), suggesting greater uncertainty in that parameter. Notably, the indirect effect has perfect coverage (1.00) and a small bias (0.00214), reflecting reliable estimation of the mediation pathway, although the total effect coverage (0.86) falls just below the typical target for credible intervals. Overall, these findings support the stability and accuracy of the Bayesian mediation model, while highlighting areas where parameter estimates may benefit from increased sample size, more precise priors, or further model refinement.

## Discussion

Mediation analysis has emerged as a powerful approach for disentangling complex causal pathways in patient-reported outcomes (PROs), particularly in clinical contexts where traditional methods fall short due to nonlinearity, missing data, or the need to model multiple mediators. For example, Lindmark and Darehed (2025) [13] applied causal mediation analysis to a large national stroke registry to quantify how socioeconomic disparities in PROs—such as fatigue, pain, and general health—could be reduced through targeted interventions on modifiable mediators like smoking and metabolic health. Their findings underscore the utility of mediation models in identifying intervention targets that meaningfully influence patient-perceived outcomes. Yu et al. (2023) [14] similarly demonstrated the application of Bayesian mediation methods to investigate racial and ethnic disparities in anxiety among cancer survivors, highlighting the framework's robustness to prior specification and its capacity for hierarchical modeling. Beyond mediation, Bayesian network approaches have also gained traction for optimizing PRO collection and analysis. Yücetürk et al. (2022) [15] used Bayesian networks to reduce the question burden in PROMs by adaptively selecting the most informative items, improving efficiency without sacrificing measurement accuracy. Greco et al. (2024) [16] further leveraged Bayesian network theory to reveal dependencies among treatment, spasticity severity, and quality-of-life indicators in multiple sclerosis trials, effectively identifying causal pathways and mediation effects. Collectively, these studies illustrate the versatility of Bayesian methods in advancing the analysis of PROs, particularly in digital and AI-driven settings, where data are often high-dimensional, adaptive, and rich in contextual detail.

This study highlights the transformative potential of integrating artificial intelligence (AI) chatbots into clinical trials for collecting patient-reported outcomes (PROs) and advancing causal inference through Bayesian mediation analysis. By leveraging AI chatbots, we were able to gather real-time, interactive data on fatigue, stress, and adverse effects while ensuring scalability and accessibility. The use of GPT-4 in simulating patient-chatbot dialogues enabled the generation of a robust synthetic dataset, which provided a valuable framework for testing and validating Bayesian mediation models.

The findings demonstrated that Bayesian mediation analysis is a powerful tool for disentangling direct and indirect effects in causal pathways, particularly in complex clinical scenarios. The low bias, robust coverage, and reliable estimation of key parameters (e.g., indirect effects) observed in this study underscore the robustness of the Bayesian approach. Notably, the model's ability to incorporate prior information and quantify uncertainty through credible intervals makes it especially suitable for analyzing real-world data where complexities such as missing data and small sample sizes are common.

However, the results also identified areas for improvement. The parameter representing the effect of the mediator on the outcome, exhibited a larger root mean square error (RMSE) and slightly lower coverage, suggesting potential variability in the indirect pathway estimates. Additionally, the total effect coverage was marginally below the conventional threshold, indicating that refining model specifications or increasing the sample size may further improve estimation accuracy. Future work could explore the impact of more precise priors, larger datasets, or alternative modeling approaches to address these limitations.

These Bayesian frameworks are particularly relevant to the future integration of AI chatbot–collected patient feedback into clinical trial analyses. As clinical trials increasingly incorporate real-time digital engagement tools to monitor symptoms, adherence, and patient experiences, Bayesian mediation models could be leveraged to rigorously analyze PRO data obtained via chatbots, accounting for complex mediator relationships and missing data patterns. The ability to flexibly model nonlinear and mixed-type variables, as demonstrated in Bayesian mediation frameworks [17], would be especially valuable in this context where chatbot interactions may yield high-dimensional, unstructured, and time-varying data. Applying Bayesian mediation analysis to AI-driven PRO collection could thus improve our understanding of factors influencing patient retention, withdrawal, and trial outcomes, and may ultimately support more patient-centered, adaptive trial designs.

Looking forward, several directions can further extend the utility and robustness of this framework. First, incorporating individual-level patient characteristics—such as age, comorbidities, medication history, or psychosocial variables—into the Bayesian mediation model would allow for the investigation of effect modification and improve the personalization of causal inferences. These covariates can be integrated either as additional predictors of the mediator and outcome or as interaction terms that condition the strength of mediation pathways. Second, future work should consider modeling the variability inherent in chatbot interactions themselves. Since different prompt templates or branching logic structures may influence the types of patient responses captured, incorporating chatbot prompt variability as an additional layer in a hierarchical Bayesian model may help account for systematic differences across templates, versions, or usage contexts. Such extensions would enhance the generalizability of the mediation framework and improve its adaptability to real-world deployments where chatbot behavior may evolve over time or vary across populations. Together, these advancements would strengthen the capacity of AI-driven tools to support rigorous, scalable, and personalized causal inference in clinical research.

The incorporation of AI chatbots in clinical trials not only streamlines PRO data collection but also enhances the granularity and contextual richness of the data. By capturing both structured responses and unstructured patient narratives, chatbots provide a comprehensive dataset for identifying nuanced cause-and-effect relationships. Furthermore, their adaptability and real-time functionality offer significant advantages in monitoring patient symptoms, detecting early warning signs, and delivering personalized interventions.

This study underscores the importance of combining AI-driven data collection with advanced analytical techniques like Bayesian mediation analysis to maximize the value of clinical trial data. The approach presented here has broad implications for improving patient engagement, refining causal inferences, and advancing personalized healthcare. As the adoption of AI chatbots continues to grow, future research should explore their integration into diverse clinical contexts and evaluate their long-term impact on patient outcomes and healthcare efficiency.

## Author contributions

**Conceptualization:** Shihao Shen, Jun Yin.

**Formal analysis:** Shihao Shen.

**Methodology:** Shihao Shen, Jun Yin.

**Software:** Shihao Shen.

**Visualization:** Shihao Shen.

**Writing – original draft:** Shihao Shen, Jun Yin.

**Writing – review & editing:** Shihao Shen, Jun Yin.

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
