## [Decision Letter · Decision Letter 0]

Dear Dr. Shen,

Thank you for submitting your manuscript to PLOS ONE. After careful consideration, we feel that it has merit but does not fully meet PLOS ONE’s publication criteria as it currently stands. Therefore, we invite you to submit a revised version of the manuscript that addresses the points raised during the review process.

The reviewers commended the novelty and technical soundness of your work. However, to enhance clarity and scholarly rigour, there is a need to revise the abstract to include research gaps, results, and a well-crafted conclusion. Strengthen the introduction by clearly stating the research objectives and questions. Enhance your discussion with supporting literature, disclose study limitations, and suggest future research. Finally, enrich your reference list with recent sources. We look forward to receiving your revised manuscript addressing these minor but important points. Thank you.

Musa Adekunle Ayanwale

Academic Editor

PLOS ONE

We look forward to receiving your revised manuscript.

Kind regards,

Musa Adekunle Ayanwale

Academic Editor

PLOS ONE

Reviewers' comments:

Reviewer's Responses to Questions

**Comments to the Author**

1. Is the manuscript technically sound, and do the data support the conclusions?

Reviewer #1: Yes

Reviewer #2: Yes

2. Has the statistical analysis been performed appropriately and rigorously?

Reviewer #1: Yes

Reviewer #2: Yes

3. Have the authors made all data underlying the findings in their manuscript fully available?

Reviewer #1: Yes

Reviewer #2: Yes

4. Is the manuscript presented in an intelligible fashion and written in standard English?

Reviewer #1: Yes

Reviewer #2: Yes

Reviewer #1: Your title is strong but could be clearer and more precise. Suggested revision: "Bayesian Mediation Analysis Using Patient-Reported Outcomes from AI Chatbots to Infer Causal Pathways in Clinical Trials."

The abstraction is well-structured and technically sound, but slight refinements could improve clarity and conciseness. For example, rewording "linking treatment effects to outcomes via mediators such as adverse events and covariates" to "linking treatment effects to outcomes through mediators like adverse events and patient-specific covariates" enhances flow. Additionally, specifying whether "synthetic patient-chatbot dialogues were generated to validate the Bayesian mediation framework" were used solely for validation or also for model training/testing would improve precision. Lastly, simplifying "underscoring the model's potential to enhance the granularity and accuracy of clinical trial data analysis" to "underscoring its potential to improve clinical trial data accuracy and depth" makes the sentence more concise while retaining meaning.

Include a theoretical framework in your study.

The materials and methods are clear and well-structured, but minor corrections can improve accuracy and readability.

Would you like to expand on the potential sources of variability in b beyond sample size and model specification, such as differences in patient characteristics or chatbot interaction nuances?

References are scanty; rework.

Reviewer #2: It is a great pleasure to have reviewed this paper, "Bayesian Mediation Analysis Using Patient Reported Outcome Collected from AI Chat to Estimate Causal Pathways in Clinical Trials". The study has numerous merits, but for the optic of this review, I will comment on some grey areas that, after implementing the correction, will further strengthen the quality of the paper.

Abstract

The authors fail to address the study gap. The gap in the study lies in the lack of detailed analysis of the variables that may have influenced the result of absence. Results were absent in the abstract, and the conclusion was not well crafted. Additionally, the conclusion fails to summarize the key findings and implications of the study effectively.

Introduction

The authors should identify the research objectives and discuss the objective variables in the introduction session. This will provide readers with a clear understanding of the purpose of the study and the specific factors that will be analyzed. By clearly outlining the research objectives and variables, the authors can set the stage for the rest of the paper and guide readers through the methodology and results logically and coherently. Additionally, clearly defining the objectives and variables can help ensure the study stays focused and addresses the key questions or hypotheses being investigated.

The authors fail to raise any research questions addressed in the study, leaving readers wondering about the significance of their findings. Without straightforward research questions, it is difficult to understand the purpose of the study and how the results contribute to existing knowledge in the field. In future research, it is essential for authors to clearly define their research questions to guide the study and provide context for their findings.

Materials and methods

Authors should provide more information on how the study was replicated, including details on the sample size, data collection methods, and statistical analyses used. This level of transparency is essential for both peer review and future replication attempts, as it allows other researchers to assess the validity and reliability of the findings accurately. Additionally, authors should clearly outline any limitations or potential biases in their study design and any steps taken to mitigate these challenges. By providing a detailed account of the replication process, authors can contribute to advancing scientific knowledge and ensure that their research is rigorously examined and verified.

Discussion

The authors fail to support discussion sessions with other studies, limiting their findings' credibility. Without additional research to back up their claims, it isn't easy to fully trust the conclusions drawn in the study. For this research to be impactful and influential in the field, the author must incorporate more thorough comparisons and analysis with other relevant studies. Only then can the study be considered comprehensive and reliable.

Limitations and suggestions for further studies

The authors should disclose the study limitations and suggestions further studies to provide a more comprehensive understanding. By acknowledging the constraints of their research, such as sample size or data collection methods, the authors can help readers interpret the results more accurately. Additionally, offering ideas for future research directions can inspire other scholars to explore new aspects of the subject and build upon the existing findings. Overall, transparent communication regarding study limitations and recommendations for future studies is essential for advancing knowledge in the field.

References

More recent authors need to be cited and referred to accurately represent the current state of knowledge on the topic. It is essential to acknowledge the contributions of newer voices in the field and to give credit where credit is due. By including references to contemporary authors, researchers can demonstrate that they are up-to-date with the latest research and theories in their study area. This adds credibility to their work and helps advance the conversation and understanding of the topic.

**Do you want your identity to be public for this peer review?** For information about this choice, including consent withdrawal, please see our Privacy Policy

Reviewer #1: **Yes: ** Olajumoke Olayemi Salami

Reviewer #2: No

---

## [Author Response · Author response to Decision Letter 1]

4 May 2025

Reviewer #1:

C1: Your title is strong but could be clearer and more precise. Suggested revision: "Bayesian Mediation Analysis Using Patient-Reported Outcomes from AI Chatbots to Infer Causal Pathways in Clinical Trials."

Response: The title has been updated accordingly. Thanks for the comment.

C2: The abstraction is well-structured and technically sound, but slight refinements could improve clarity and conciseness. For example, rewording "linking treatment effects to outcomes via mediators such as adverse events and covariates" to "linking treatment effects to outcomes through mediators like adverse events and patient-specific covariates" enhances flow. Additionally, specifying whether "synthetic patient-chatbot dialogues were generated to validate the Bayesian mediation framework" were used solely for validation or also for model training/testing would improve precision. Lastly, simplifying "underscoring the model's potential to enhance the granularity and accuracy of clinical trial data analysis" to "underscoring its potential to improve clinical trial data accuracy and depth" makes the sentence more concise while retaining meaning.

Response: The abstract has been updated according to the comment. As part of the updates, I have clarified that “Synthetic patient-chatbot dialogues were generated to evaluate the performance of the Bayesian mediation framework”. The simulated dialogues were used for validation only, not for fine-tuning to improve precision.

C3: Include a theoretical framework in your study.

Response: We have added a paragraph about the theoretical framework of the Bayesian mediation model:

Based on the parametric Bayesian mediation framework, mediator model is

M_i~Normal(i_M +a ∙X_i,σ_M)

And the outcome model is

y_i~Normal(i_y +c^' ∙X_i+d ∙Z_i+b ∙M_i,σ_Y)

where i indexes each observation, i_M and i_y are intercept parameters. Note that X_i, Z_i, M_i and〖 Y〗_i are observed data.

The direct effect given by c′ is the portion of the effect of the independent variable on the outcome that is not mediated by the mediator. The indirect effect given by a•b is the portion of the effect that is mediated through the mediator. The total effect given by c′ + a•b is the sum of the direct and indirect effects.

C4: The materials and methods are clear and well-structured, but minor corrections can improve accuracy and readability.

Response: We have made the following edits in the materials and methods section to improve readability.

To clarify chatbot design and routing logic in Figure 1, we have revised the description about: “We developed an AI chatbot to collect PROs related to fatigue, stress, and adverse effects using a structured conversation flow illustrated in Figure 1.”

To clarify the chatbot dialogue process, we have revised the following statement about simulation procedure: “To simulate chatbot–user interactions, we used GPT-4o to alternate between the roles of “agent” (the AI chatbot) and “user” (the patient), allowing generation of realistic, bidirectional conversations. We began by defining a set of representative patient concerns related to fatigue, stress, and side effects. These scenarios guided the chatbot’s scripted decision logic, which was implemented through GPT-4o prompts to simulate realistic conversational pathways.”

To improve model description consistency, we have also revised the following statement: “As shown in Figure 2, we applied a Bayesian mediation framework to quantify the direct effect of treatment (X) on fatigue (Y), and the indirect effect mediated by adverse events (M), with stress (Z) included as a covariate influencing the outcome.”

C5: Would you like to expand on the potential sources of variability in b beyond sample size and model specification, such as differences in patient characteristics or chatbot interaction nuances?

Response: It is a great question. We are also considering future works in expanding the framework to account for the patient characteristics by including patient characteristics as covariates; or to account for the variability of chatbot prompt templates by another layers in the hierarchical modeling. Although these additional features will be out of the scope of this manuscript, we have added a paragraph in the discussion section about these future directions:

“Looking forward, several directions can further extend the utility and robustness of this framework. First, incorporating individual-level patient characteristics—such as age, comorbidities, medication history, or psychosocial variables—into the Bayesian mediation model would allow for the investigation of effect modification and improve the personalization of causal inferences. These covariates can be integrated either as additional predictors of the mediator and outcome or as interaction terms that condition the strength of mediation pathways. Second, future work should consider modeling the variability inherent in chatbot interactions themselves. Since different prompt templates or branching logic structures may influence the types of patient responses captured, incorporating chatbot prompt variability as an additional layer in a hierarchical Bayesian model may help account for systematic differences across templates, versions, or usage contexts. Such extensions would enhance the generalizability of the mediation framework and improve its adaptability to real-world deployments where chatbot behavior may evolve over time or vary across populations. Together, these advancements would strengthen the capacity of AI-driven tools to support rigorous, scalable, and personalized causal inference in clinical research.”

C6: References are scanty; rework.

Response: We have expanded the References and Discussion section to include citations of recent studies that demonstrate the evolving utility of mediation analyses on the patient-reported outcomes, across diverse clinical and population health contexts.

“Mediation analysis has emerged as a powerful approach for disentangling complex causal pathways in patient-reported outcomes (PROs), particularly in clinical contexts where traditional methods fall short due to nonlinearity, missing data, or the need to model multiple mediators. For example, Lindmark and Darehed (2025) [1] applied causal mediation analysis to a large national stroke registry to quantify how socioeconomic disparities in PROs—such as fatigue, pain, and general health—could be reduced through targeted interventions on modifiable mediators like smoking and metabolic health. Their findings underscore the utility of mediation models in identifying intervention targets that meaningfully influence patient-perceived outcomes. Yu et al. (2023) [2] similarly demonstrated the application of Bayesian mediation methods to investigate racial and ethnic disparities in anxiety among cancer survivors, highlighting the framework’s robustness to prior specification and its capacity for hierarchical modeling. Beyond mediation, Bayesian network approaches have also gained traction for optimizing PRO collection and analysis. Yücetürk et al. (2022) [3] used Bayesian networks to reduce the question burden in PROMs by adaptively selecting the most informative items, improving efficiency without sacrificing measurement accuracy. Greco et al. (2024) [4] further leveraged Bayesian network theory to reveal dependencies among treatment, spasticity severity, and quality-of-life indicators in multiple sclerosis trials, effectively identifying causal pathways and mediation effects. Collectively, these studies illustrate the versatility of Bayesian methods in advancing the analysis of PROs, particularly in digital and AI-driven settings, where data are often high-dimensional, adaptive, and rich in contextual detail.”

We hope these additions address your recommendation and further support the methodological relevance of our approach.

Reviewer #2:

It is a great pleasure to have reviewed this paper, "Bayesian Mediation Analysis Using Patient Reported Outcome Collected from AI Chat to Estimate Causal Pathways in Clinical Trials". The study has numerous merits, but for the optic of this review, I will comment on some grey areas that, after implementing the correction, will further strengthen the quality of the paper.

C1: Abstract

The authors fail to address the study gap. The gap in the study lies in the lack of detailed analysis of the variables that may have influenced the result of absence. Results were absent in the abstract, and the conclusion was not well crafted. Additionally, the conclusion fails to summarize the key findings and implications of the study effectively.

Response: We have revised the abstract to further highlight the current gap of lacking of frameworks for analyzing the AI chatbot datasets; and emphasize the impact of the study on analyzing the mediators of the patient-reported outcomes of fatigue, by applying a Bayesian mediation framework to PROs collected via AI chatbot interactions.

We have also incorporated these changes above to the changes suggested by the first reviewer in his comment about the abstract, starting from “The abstraction is well-structured and technically sound, but slight refinements could improve clarity …”. Below is the updated section:

“The integration of artificial intelligence (AI) chatbots into clinical trials offers a transformative approach to collecting patient-reported outcomes (PROs). Despite the increasing use of AI chatbots for real-time, interactive data gathering, systematic frameworks for analyzing these rich datasets—especially in uncovering causal relationships—remain limited. This study addresses this gap by applying a Bayesian mediation framework to PROs collected via AI chatbot interactions, uncovering causal pathways linking treatment effects to outcomes through mediators like adverse events and patient-specific covariates.”

We have also added details about the results in the abstract and highlight the implication of this study in the revised abstract:

“The results demonstrated low bias (<0.05), robust coverage (>85%), in estimation of the direct, indirect effect and other variables of the mediation pathways. By integrating AI chatbot-based PRO collection with Bayesian mediation analysis, this study presents a scalable and adaptive framework for quantifying causal pathways, enhancing the quality of patient-reported data, and supporting personalized, data-driven decision-making in clinical trials."

C2: Introduction

The authors should identify the research objectives and discuss the objective variables in the introduction session. This will provide readers with a clear understanding of the purpose of the study and the specific factors that will be analyzed. By clearly outlining the research objectives and variables, the authors can set the stage for the rest of the paper and guide readers through the methodology and results logically and coherently. Additionally, clearly defining the objectives and variables can help ensure the study stays focused and addresses the key questions or hypotheses being investigated.

The authors fail to raise any research questions addressed in the study, leaving readers wondering about the significance of their findings. Without straightforward research questions, it is difficult to understand the purpose of the study and how the results contribute to existing knowledge in the field. In future research, it is essential for authors to clearly define their research questions to guide the study and provide context for their findings.

Response: Thanks for the comments. We have added a paragraph in the introduction section to further discuss the research objectives, variables and research questions:

“The objective of this study is to evaluate the application of a Bayesian mediation framework to analyze patient-reported outcomes (PROs) collected through AI chatbot interactions in a clinical research setting. Specifically, we seek to address the research questions of (1) whether the Bayesian mediation framework can reliably estimate direct and indirect effects using PRO data gathered from chatbot conversations, and (2) we aim to demonstrate how causal pathways between treatment exposure and patient outcomes can be uncovered using chatbot-collected data. In this framework, the primary exposure variable is the treatment assignment (active vs. placebo), the primary outcome is patient-reported fatigue, and the mediator is the occurrence of adverse events as reported during the chatbot dialogue. Additionally, stress-related factors reported by patients are incorporated as covariates influencing the outcome. By modeling these relationships, we seek to provide a novel and effective strategy for leveraging AI-driven PRO data to enhance causal inference and refine understanding of treatment effects in clinical trials.”

C3: Materials and methods

Authors should provide more information on how the study was replicated, including details on the sample size, data collection methods, and statistical analyses used. This level of transparency is essential for both peer review and future replication attempts, as it allows other researchers to assess the validity and reliability of the findings accurately. Additionally, authors should clearly outline any limitations or potential biases in their study design and any steps taken to mitigate these challenges. By providing a detailed account of the replication process, authors can contribute to advancing scientific knowledge and ensure that their research is rigorously examined and verified.

Response: The details of the simulation approach were provided both in the methods section about the chatbot conversations that were simulated with the flow-chart diagram in the Figure 1 and in the first paragraph of the results section: about the 100 iterations of the simulations were conducted, each with 75 subjects randomized 1:1 into the active or placebo groups, followed by the simulation results and explanations of the variables.

To clarify the chatbot dialogue process, we have revised the following statement about simulation procedure: “To simulate chatbot–user interactions, we used GPT-4o to alternate between the roles of “agent” (the AI chatbot) and “user” (the patient), allowing generation of realistic, bidirectional conversations. We began by defining a set of representative patient concerns related to fatigue, stress, and side effects. These scenarios guided the chatbot’s scripted decision logic, which was implemented through GPT-4o prompts to simulate realistic conversational pathways.”

An additional paragraph about the Bayesian mediation framework was added:

“Based on the parametric Bayesian mediation framework, mediator model is

M_i~Normal(i_M +a ∙X_i,σ_M)

And the outcome model is

y_i~Normal(i_y +c^' ∙X_i+d ∙Z_i+b ∙M_i,σ_Y)

where i indexes each observation, i_M and i_y are intercept parameters. Note that X_i, Z_i, M_i and〖 Y〗_i are observed data.

The direct effect given by c′ is the portion of the effect of the independent variable on the outcome that is not mediated by the mediator. The indirect effect given by a•b is the portion of the effect that is mediated through the mediator. The total effect given by c′ + a•b is the sum of the direct and indirect effects.”

More details about the limitation of the approach were also added in the manuscript. While this study demonstrates the feasibility of applying a Bayesian mediation framework to PRO data collected through AI chatbot interactions, several methodological limitations should be noted. First, although our Bayesian mediation model accounts for key mediators, it does not fully incorporate a broader range of individual patient characteristics—such as demographic, clinical, or psychosocial factors—that could influence both mediator and outcome relationships. Future extensions of the framework could integrate multiple patient-specific covariates to more precisely model heterogeneity in causal pathways. Second, the current simulation approach assumes a standardized chatbot dialogue structure; however, in real-world applications, the variability of chatbot–patient conversations—arising from diverse communication styles, language nuances, and interaction patterns—could introduce additional uncertainty and complexity. Adapting the

---

## [Decision Letter · Decision Letter 1]

Bayesian Mediation Analysis Using Patient-Reported Outcomes from AI Chatbots to Infer Causal Pathways in Clinical Trials

PONE-D-25-04459R1

Dear Dr. Shihao,

We’re pleased to inform you that your manuscript has been judged scientifically suitable for publication and will be formally accepted for publication once it meets all outstanding technical requirements.

Kind regards,

Musa Adekunle Ayanwale

Academic Editor

PLOS ONE

Additional Editor Comments (optional):

The authors have thoroughly addressed all the initial concerns raised by the reviewers. Thank you for your careful revisions.

Reviewers' comments:

Reviewer's Responses to Questions

**Comments to the Author**

Reviewer #1: (No Response)

Reviewer #2: All comments have been addressed

2. Is the manuscript technically sound, and do the data support the conclusions?

Reviewer #1: Yes

Reviewer #2: Yes

3. Has the statistical analysis been performed appropriately and rigorously?

Reviewer #1: Yes

Reviewer #2: Yes

4. Have the authors made all data underlying the findings in their manuscript fully available?

Reviewer #1: Yes

Reviewer #2: Yes

5. Is the manuscript presented in an intelligible fashion and written in standard English?

Reviewer #1: Yes

Reviewer #2: Yes

Reviewer #1: The author needs to write a literature review for this study, incorporating relevant and up-to-date scholarly sources.

Minor grammatical adjustments for better readability.

The references are too limited; please add more relevant sources.

Reviewer #2: (No Response)

**Do you want your identity to be public for this peer review?** For information about this choice, including consent withdrawal, please see our Privacy Policy

Reviewer #1: **Yes: ** Olajumoke Olayemi Salami

Reviewer #2: **Yes: ** Oluwaseyi Aina Gbolade Opesemowo

---

## [Editor Report · Acceptance letter]

PONE-D-25-04459R1

PLOS ONE

Dear Dr. Yin,

I'm pleased to inform you that your manuscript has been deemed suitable for publication in PLOS ONE. Congratulations! Your manuscript is now being handed over to our production team.

Kind regards,

on behalf of

Dr Musa Adekunle Ayanwale

Academic Editor

PLOS ONE